# The deep conservation of the Lepidoptera Z chromosome suggests a non-canonical origin of the W

Christelle Fraïsse[1], Marion A.L. Picard[1] & Beatriz Vicoso[1]

Moths and butterflies (Lepidoptera) usually have a pair of differentiated WZ sex chromosomes. However, in most lineages outside of the division Ditrysia, as well as in the sister order Trichoptera, females lack a W chromosome. The W is therefore thought to have been acquired secondarily. Here we compare the genomes of three Lepidoptera species (one Dytrisia and two non-Dytrisia) to test three models accounting for the origin of the W: (1) a Z-autosome fusion; (2) a sex chromosome turnover; and (3) a non-canonical mechanism (e.g., through the recruitment of a B chromosome). We show that the gene content of the Z is highly conserved across Lepidoptera (rejecting a sex chromosome turnover) and that very few genes moved onto the Z in the common ancestor of the Ditrysia (arguing against a Z-autosome fusion). Our comparative genomics analysis therefore supports the secondary acquisition of the Lepidoptera W by a non-canonical mechanism, and it confirms the extreme stability of well-differentiated sex chromosomes.

[1] Institute of Science and Technology Austria, Am Campus 1, Klosterneuburg 3400, Austria. Christelle Fraïsse and Marion A. L. Picard contributed equally to this work. Correspondence and requests for materials should be addressed to B.V. (email: beatriz.vicoso@ist.ac.at)

Despite the near-ubiquity of sexual reproduction in animals and the partial conservation of pathways involved in sex determination, differentiated sex chromosomes have evolved independently in many clades[1]. Chromosomal sex determination can take two forms: male-heterogamety (XY systems, such as those of mammals) and female heterogamety (females are WZ, males ZZ, such as in birds); in both cases the sex-specific chromosome (Y or W) is often gene poor and consists mostly of repetitive DNA. Under the canonical model, such differentiated sex chromosomes initially arise from standard pairs of autosomes: after acquiring a sex-determining gene, newly formed Y (or W) chromosomes accumulate sexually antagonistic mutations that are favorable to the sex they are found in (reviewed in Wright et al.[2]). The presence of such mutations eventually favours the loss of recombination between the sex chromosomes, in order to achieve full linkage between sex and alleles with sex-specific benefits, and sets in motion the degeneration of the sex-specific chromosome[2]. Additional autosomes can further be added to the sex chromosome component through fusions to either sex chromosome, and become similarly differentiated. Direct support for a shared autosomal origin of the X and Y chromosomes (and Z and W chromosomes) is found in several vertebrate groups, where Y/W-linked genes are largely homologous to X/Z-linked ones[3]. This evidence is lacking in other clades, such as *Drosophila* or Lepidoptera (moths and butterflies), in which no consistent homology is detected between X/Z and Y/W-linked genes[4, 5].

All butterflies and most moths, including the model silkworm (*Bombyx mori*, Bombycidae) and all species for which there is extensive linkage information, belong to the division Ditrysia. The vast majority of what is known about Lepidoptera sex chromosome evolution concerns only this group, which shares a heterologous

**Fig. 1** Evidence and models for the secondary origin of the W chromosome. **a** Presence/absence of the W chromosome across a simplified phylogeny of Lepidoptera. The table shows, for each family, the total number of chromosomes; the presence (yes), absence (no) or non-determination (n.d.) of sex chromatin; as well as the cytogenetic confirmation of the presence of the W chromosome (adapted from Dalíková et al. 15). In Hepialidae, only some species, which did not include *T. sylvina*, were cytogenetically investigated to assess the presence/absence of the W (asterisk). Dotted lines represent families that are not investigated in our study, while gray circles show the five studied species. **b** Hypotheses for the secondary acquisition of the W chromosome. The ancestral Z0 female karyotype is represented by the Z chromosome (in red) and two pairs of autosomes (A1 and A2, in blue). In the two canonical models (Z-autosome fusion and sex chromosome turnover), the new sex chromosomes derived from a standard pair of autosomes (A1). In the non-canonical model, the W chromosome has a non-autosomal origin (in black), e.g., the recruitment of a B chromosome. The blue dotted lines represent the secondary degradation of the neo-W chromosome

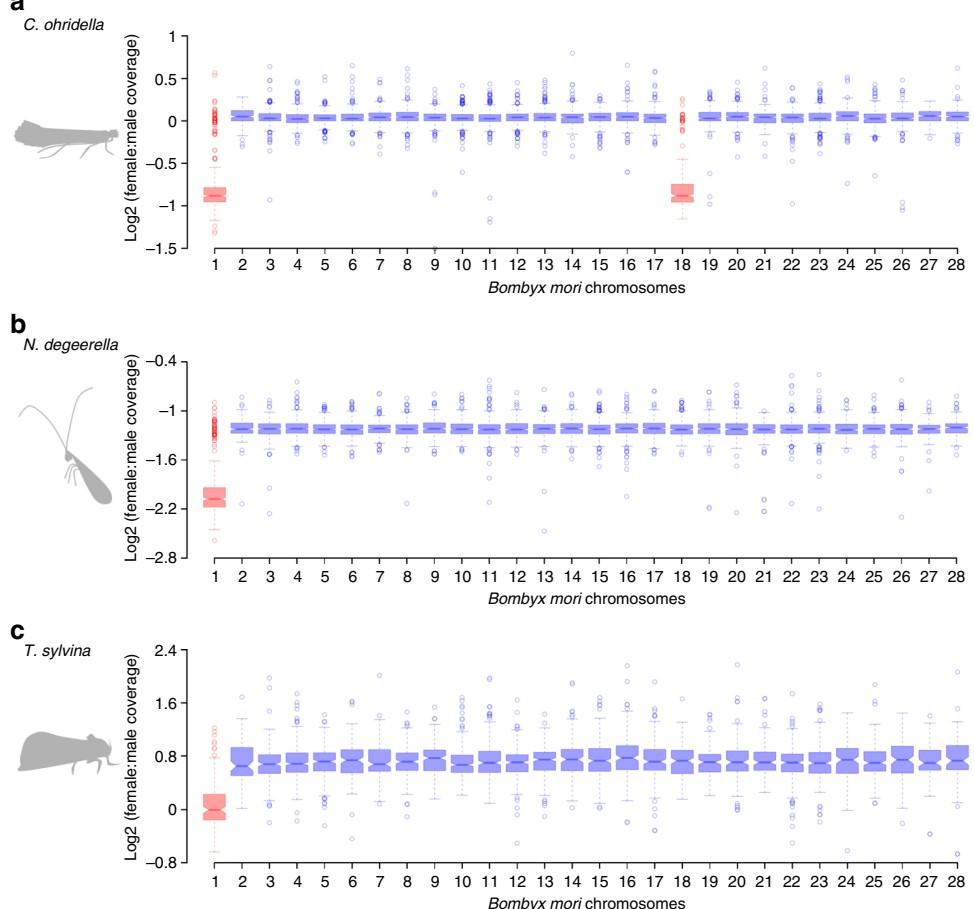

**Fig. 2** The Z chromosome of *B. mori* is homologous to that of the other species. *Cameraria ohridella* has additionally acquired a neo-Z chromosome (Chr. 18). For each species, scaffolds were assigned to one of the *B. mori* chromosomes based on their gene content. The Log2 of the female to male coverage ratio of each chromosome is shown for **a** *C. ohridella*, **b** *N. degeerella*, and **c** *T. sylvina*. Scaffolds mapping to the *B. mori* Z chromosome, and the neo-Z chromosome of *C. ohridella*, are in red, those mapping to the autosomes are in blue

pair of WZ sex chromosomes. The gene content of the Z chromosome is well conserved even between distant species within the Ditrysia[6], although large-scale chromosomal rearrangements involving the Z chromosome have occurred in some lineages (Tortricidae[7], Saturnidae[8], and Nymphalidae[9]). The history of the W chromosome is hard to assess even within Ditrysia, as the absence of protein coding genes on this chromosome[5] makes comparisons between different species difficult and prevents a direct test of homology with the Z chromosome. However, extensive cytogenetic analyses have shown that both the gain and loss of W chromosomes are frequent in Lepidoptera, with closely related species often showing differences in their sex chromosome number[10].

Since Lepidoptera and Trichoptera are sister orders and two major insect lineages with female heterogamety, this system is thought to have originated in their common ancestor. Interestingly, both Trichoptera and at least one species of Micropterigidae, the most distantly related lineage to Ditrysia within the Lepidoptera, are Z0: females only carry one Z chromosome and lack a W chromosome. In Ditrysia, W chromosomes can usually be detected in interphase spreads as female-specific heterochromatin bodies. The absence of such sex chromatin bodies in other independent non-ditrysian lineages[11] has led to the suggestion that the Z0 system is ancestral and that the W chromosome was acquired secondarily either in the ancestor of all Ditrysia and Tischerioidea[11–14], or early in the evolution of the Ditrysia[15] (Fig. 1a).

Several models could account for such a secondary acquisition of the W (Fig. 1b), of which two have been favoured[11, 14]: (i) a Z-autosome fusion; (ii) the recruitment of a B chromosome (a dispensable chromosome found in only some individuals of a population) to perform female-specific functions. While such a recruitment of B chromosomes for sex determination has been proposed for several systems (e.g., *Drosophila*[16], Homoptera[17]), it has not been empirically demonstrated. A third model that has so far not been considered is sex chromosome turnover: if the ancestral Z reverted to an autosome and a new sex chromosome pair took over sex determination, this would also result in a neo-W chromosome (Fig. 1b). Such turnover of ancestral sex chromosomes is unusual, but was recently found to have occurred repeatedly in flies and mosquitoes[18]. In the absence of W-linked genes in Lepidoptera, testing these different models requires detailed comparisons of the gene content of the Z chromosome of Z0 non-Ditrysia and WZ Ditrysia, something that is still lacking.

To investigate the origin of the W, we analyze the genomes of three Lepidoptera species: *Cameraria ohridella* (the horse-chestnut leaf miner, family Gracillariidae) is a ditrysian lineage that has a WZ karyotype[19], while *Nemophora degeerella* (a fairy longhorn moth, family Adelidae), and *Triodia sylvina* (the orange swift, family Hepialidae) represent two independent non-ditrysian lineages that are presumed to be lacking a W due to the absence of sex chromatin[11, 12]. We compare male and female genomic coverage to detect Z-derived scaffolds and show that the

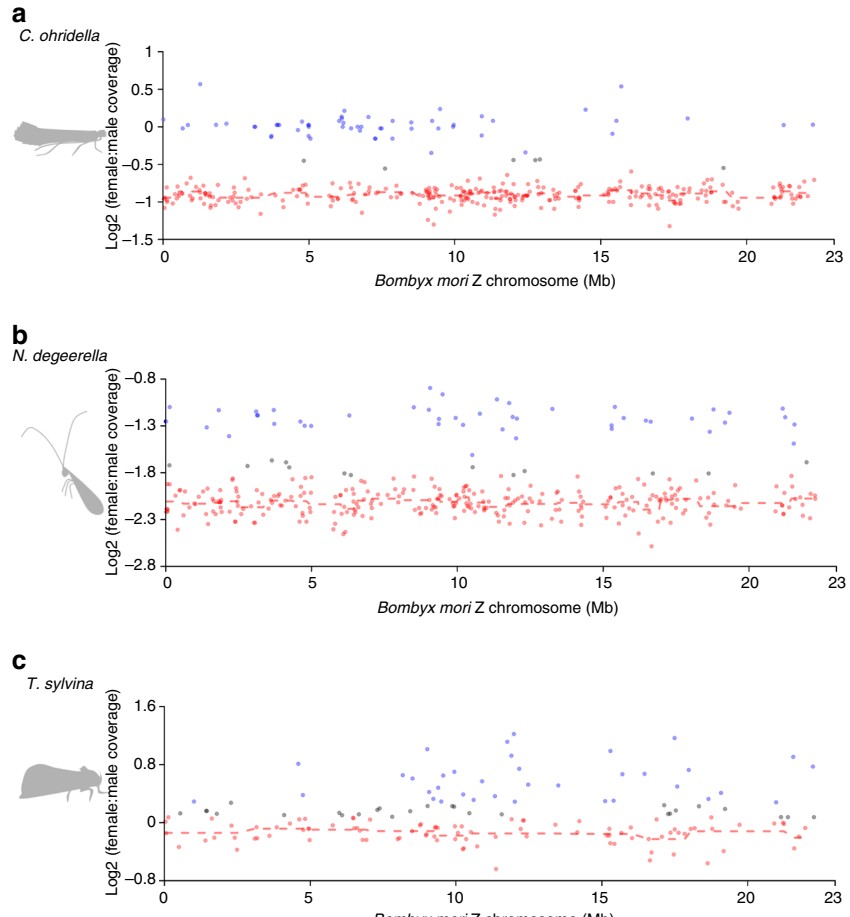

**Fig. 3** Homology between the Z chromosome of *B. mori* and that of the other species. For each species, scaffolds mapping to the Z chromosome of *B. mori* (Chr. 1) have been classified as Z-linked (in red), autosomal (in blue), or unclassified (in gray). The Log2 of the female to male coverage ratio along the *B. mori* Z chromosome (Chr. 1) is shown for **a** *C. ohridella*, **b** *N. degeerella*, and **c** *T. sylvina*. Dashed lines are moving average based on sliding windows of 10 scaffolds

Z chromosome of the different species is homologous to the Z chromosome of the model *B. mori*[20] and that *C. ohridella* has further acquired a neo-Z chromosome (Chr. 18 of *B. mori*). In order to test for a Z-autosome fusion at the base of the Ditrysia, which could account for the appearance of the W, we investigate gene movement onto and off the Z chromosome. We show that very few genes moved onto the Z in the common ancestor of the Ditrysia, suggesting that no such fusion occurred. Finally, no female-specific sequence is identified in *N. degeerella*, consistent with a lack of W chromosome in Adelidae. These findings shed light on the evolution of the Lepidoptera Z chromosome, support the view that well-differentiated sex chromosomes are generally extremely stable, and allow us to test several theories that have been proposed for the origin of the W chromosome.

## Results

**De novo genome assemblies and genomic coverage**. We sequenced the genome of a ditrysian lineage, *C. ohridella*, and two non-ditrysian lineages that are presumed to lack a W chromosome, *N. degeerella* and *T. sylvina* (Supplementary Table 1). Scaffolds were assembled using a multi-step procedure combining SOAPdenovo2, GapCloser, SSPACE, and Cap3. Assembly results for each species are summarized in Supplementary Table 2. The final assembly consisted of 152,025 *C. ohridella* scaffolds (N50 = 6277 bp), 194,957 *N. degeerella* scaffolds (N50 = 5350 bp) and 2,353,920 *T. sylvina* scaffolds (N50 = 1184 bp). In each species,

male and female DNA reads were then mapped separately to the assemblies using Bowtie2, and genomic coverage was estimated with SOAPcoverage. Median coverages in females and males were as follow: *C. ohridella*: 60 and 60, *N. degeerella*: 49 and 117, *T. sylvina*: 22 and 13, respectively. They were generally lower in *T. sylvina* as this species has a large genome (total assembly length = 1.8 Gbp, Supplementary Table 2).

**Identification of Z-linked and autosomal genes**. Since genomic coverage is proportional to the number of copies of a gene in the genome, we compared the Log2 of the female over male (F:M) coverage to identify Z-linked and autosomal scaffolds[21] (Supplementary Data 1, Supplementary Fig. 1). In order to compare the gene content of the Z in the different species, we further mapped *B. mori* genes to the genome assemblies, and selected non-overlapping best hits to detect 1:1 orthologs for each species (*C. ohridella*: 9123, *N. degeerella*: 8359, *T. sylvina*: 5053). The proportion of Z-linked genes (genes on scaffolds with Log2 (F:M coverage) below ×0.6 the median) was similar between *B. mori* (3.7%) and *N. degeerella* (3.6%, $P = 0.098$ with a two-tailed $\chi^2$-test; Supplementary Table 3) and *T. sylvina* (2.8%, $P = 0.4393$), but higher for *C. ohridella* (5.3%, $P < 0.0001$). Using more stringent coverage limits (Supplementary Data 1, Supplementary Fig. 2), the number of unclassified genes increased from 0.2 to 7.9% in *C. ohridella*, from 0.3 to 9.4% in *N. degeerella*, and from 3.3 to 10.8% in *T. sylvina*. While the proportion of

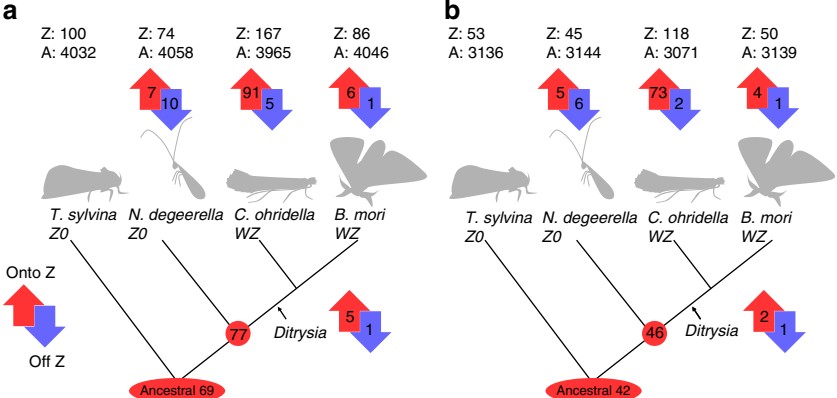

**Fig. 4** Gene movement onto and off the Z chromosome. **a** Lenient classification (4132 genes classified). **b** Stringent classification (3189 genes classified). The phylogenetic tree is adapted from Regier et al.[51]. Red circles show the number of genes present on the ancestral Z chromosome. Arrows indicate the number of genes that move onto (up red arrows) or off (down blue arrows) the Z chromosome in each lineage after the split with *T. sylvina*. The top row indicates the total number of genes classified as Z-linked (Z) or autosomal (A) in each species

Z-linked genes remained comparable in *C. ohridella* (5.6%) and *N. degeerella* (4.1%), it dropped down in *T. sylvina* (1.7%) where increased noise due to lower coverage and a fragmented assembly led many putative Z-linked scaffolds to remain unclassified.

**Conservation of the Z in Lepidoptera and Trichoptera.** Scaffolds were assigned to one of the *B. mori* chromosome based on their gene content (using the 1:1 orthologs described in the previous section; Supplementary Data 1). We could then evaluate the coverage patterns for scaffolds homologous to the different *B. mori* chromosomes, to test whether the Z chromosome of ditrysian (*B. mori* and *C. ohridella*) and non-ditrysian (*N. degeerella* and *T. sylvina*) lineages of Lepidoptera are homologous (Fig. 2). In *N. degeerella* and *T. sylvina*, scaffolds mapping to Chr. 1 (i.e., Chr. Z) of *B. mori* had a significantly lower female to male coverage ratio than scaffolds mapping to any of the autosomes of *B. mori* ($P < 0.0001$ for all comparisons, with a one-tailed Wilcoxon rank-sum test), confirming the homology of the non-ditrysian and ditrysian Z chromosomes. The same was found in *C. ohridella*, with the exception of Chr. 18, which showed patterns of coverage similar to Chr. 1 ($P = 0.423$), showing that it was additionally acquired as a neo-Z chromosome (Fig. 2a), as suggested recently[15].

The proportion of genes classified as Z-linked that also mapped to Chr. 1 was 62.1% in *C. ohridella*, 93.3% in *N. degeerella*, and 67.4% in *T. sylvina* (Supplementary Table 4). These conserved Z-linked genes were found throughout Chr. 1 (Fig. 3), rather than being restricted to a fraction of the chromosome, as might be expected for a secondary acquisition of the W in Ditrysia through a Z-autosome fusion (although such a signal would depend on the conservation of micro-synteny along the chromosome).

We extended the analysis to published female genome sequencing data from a Trichoptera (*L. lunatus*: 6646 one-to-one orthologs). In *L. lunatus*, scaffolds mapping to Chr. 1 had a significantly lower female coverage than scaffolds mapping to any autosomes ($P < 0.0001$ for all comparisons, with a one-tailed Wilcoxon rank-sum test, Supplementary Fig. 3), which further supports a shared Z chromosome between Lepidoptera and its sister lineage, Trichoptera.

**Gene movement onto and off the Z chromosome.** To further investigate whether a Z-autosome fusion occurred in the ancestor of the Ditrysia and Tischerioidea, we combined our classification of Z-linked and autosomal genes in the four Lepidoptera species and assessed gene movement onto and off the Z across the

Lepidoptera phylogeny (Fig. 4a, which is based on the 4132 orthologs that were assigned to the Z or autosomes in all species using the lenient classification; Fig. 4b, which is based on the 3189 genes classified using the stringent classification). Such a fusion should be detectable through a large number of genes moving onto the Z at the root of the Ditrysia (genes that are Z-linked in the two Ditrysia species, but not in the non-Ditrysia *T. sylvina* and *N. degeerella*). Figure 4a shows that the gene content of the Z chromosome is broadly conserved across Lepidoptera: of the 86 genes in the sample that are Z-linked in *B. mori*, 69 were present on the ancestral Z chromosome, and 77 were already Z-linked before the split of Ditrysia and *N. degeerella*. Importantly, only 5 out of 77 genes moved onto the Z at the basal branch of the Ditrysia ($P = 0.192$ when this rate is compared to the other lineages, excluding the *C. ohridella* branch, with a two-tailed $\chi^2$-test; Supplementary Table 5). Since 86 out of the 654 annotated *B. mori* Z-linked genes are represented in the sample (13%), we can rescale the tree, and approximate the total number of genes that moved onto the Z at the root of Ditrysia as 38 (the same calculation from the stringent classification yields 26 genes). Even ignoring the fact that some of these will necessarily correspond to individual gene movement onto the Z, 38 is smaller than the number of genes detected on any chromosome of the butterfly *Melitaea cinxia* (Supplementary Table. 28 of Ahola et al.[22]), which is thought to represent the ancestral ditrysian karyotype of $n = 31$[22]. Any fusion would therefore have to involve a chromosome smaller than any described so far in Ditrysia. On the other hand, we were able to detect a strong excess of genes moving onto the Z in the *C. ohridella* lineage (91 genes, $P < 0.0001$), consistent with the acquisition of a neo-Z chromosome through a Z-autosome fusion in this lineage. All our conclusions hold when using the stringent classification (Fig. 4b), or a minimal scaffold length of 500 bp instead of 1500 bp (Supplementary Fig. 4, Supplementary Table 6).

**Absence of W chromosome in *N. degeerella*.** Since the W chromosome is present only in females, we identified candidate W-linked sequences by selecting genomic scaffolds (1000 bp or more) with female-specific read coverage in each species (42 scaffolds in *C. ohridella*, two in *N. degeerella* and 2114 in *T. sylvina*). We further tested the female-specificity of all (in *N. degeerella*) and a subset (10 in *C. ohridella* and 23 in *T. sylvina*) of these W-candidates by PCR amplification in an additional three male and three female individuals (Fig. 5). In the WZ species *C. ohridella*, six of the ten tested W-candidates yielded female-

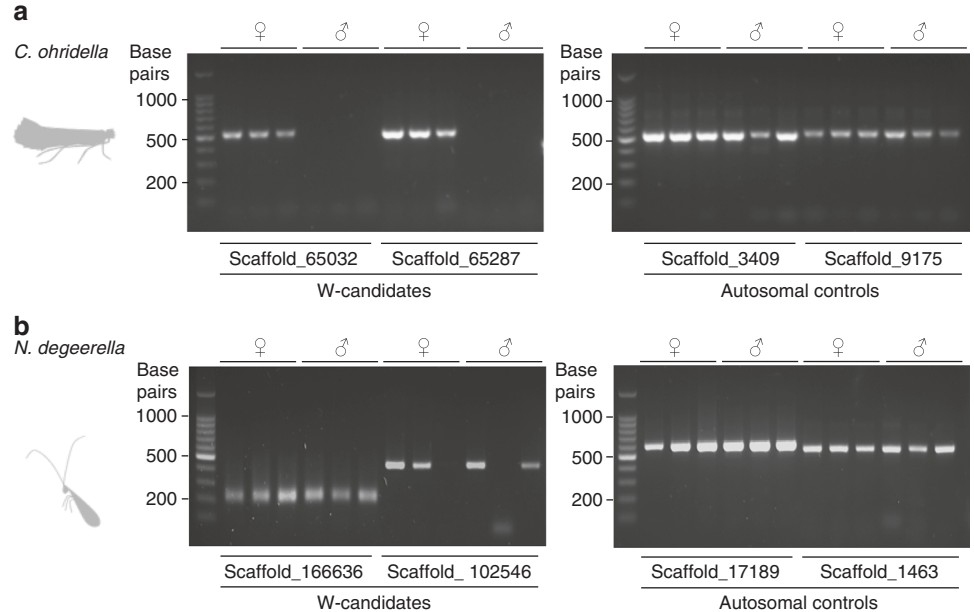

**Fig. 5** Evidence of a W chromosome in *C. ohridella* but not in *N. degeerella*. Amplification patterns of genomic DNA from three females and three males are shown for two W-candidates, and for two autosomal control genes in **a** *C. ohridella* and **b** *N. degeerella*. The molecular size marker is indicated on the left (100 bp ladder). Results for all W-candidates are shown in Supplementary Fig. 5

specific bands (Fig. 5, Supplementary Fig. 5). Contrary to this, none of the candidates of *N. degeerella* were restricted to females. In this species, we additionally tested 26 other candidates (scaffold length from 200 to 980 bp) obtained by removing the previous minimal cutoff on scaffold length (1000 bp). None were specifically amplified in females (Fig. 5, Supplementary Fig. 5). The same was found for *T. sylvina*, although in this case only 23 scaffolds (scaffold length above 2000 bp) were tested. In each of the two non-Ditrysia species, one spurious female-specific candidate was invalidated after further testing on more male and female individuals (Supplementary Fig. 5). These results support the absence of a W chromosome in *N. degeerella*. Unfortunately the poorer quality of the assembly makes such an exhaustive examination unfeasible in *T. sylvina*.

## Discussion

Previous work has shown that X and Z sex chromosomes have evolved from independent pairs of ancestral chromosomes in winged insects[23]. Our comparative genomic analysis demonstrates the conservation of the Z chromosome over 140 Myr of Lepidoptera evolution[24] and further supports a shared ancestry of the Lepidoptera and Trichoptera Z chromosomes (Supplementary Fig. 3). The Z chromosome of Lepidoptera has therefore likely been the sex-determining chromosome for 210 Myr[24], making it one of the oldest documented sex chromosome[25], and supporting the view that well-differentiated sex chromosomes are generally extremely stable, as has been reported in birds[26], lizards[27], eutherian mammals[25], and true bugs[28]. The gene content of the Z chromosome has also remained remarkably stable: 67% of the 138 *T. sylvina* genes that were assigned to the Z chromosome are also Z-linked in *B. mori*, and this number increases to 93% for *N. degeerella* (out of 300 Z-linked genes) (Supplementary Table 4). Importantly, we detect no evidence of chromosomal rearrangements involving the Z chromosome, such as Z-autosome fusions, between non-Ditrysia and Ditrysia, although such fusions have been detected in several ditrysian lineages, including *C. ohridella* (ref.[15] and this study). This is consistent with a previous analysis of karyotypes in Lepidoptera, which did not detect a decrease of the male autosome number by one pair at the base of the

ditrysian lineage[11], as would be expected if a Z-autosome fusion had occurred.

The deep conservation of the identity and gene content of the Z chromosome is at odds with the suggestion that either sex chromosome turnover or fusion of an autosome to the Z chromosome (the two 'canonical models' in Fig. 1b) has led to the acquisition of the W chromosome in the ancestor of Ditrysia and Tischerioidea, as has been proposed[11]. This leaves two possible evolutionary scenarios: (1) the W originated much earlier than currently believed, and is present in some distant non-ditrysian lineages[29]; or (2) the W has a non-canonical origin, and does not correspond to a degenerated autosome[11]. Much of the evidence for the lack of W chromosome in non-ditrysian lineages is indirect and relies on the presumed formation of sex chromatin bodies when a W chromosome is present (as is the case in Ditrysia). Such chromatin bodies are lacking in non-ditrysian lineages, except in the WZ Tischeriidae. The absence of a W chromosome has been confirmed using cytogenetics in the sister order Trichoptera[30] and in Micropterigidae, the Lepidoptera lineage that is most distantly related to Ditrysia[31]. By contrast, some species of Hepialidae have been found to carry W chromosomes despite the absence of sex chromatin[29, 32] (but ref.[29] found a $WZ_1Z_2$ karyotype, suggesting that secondary rearrangements have occurred in this clade). This is further complicated by the fact that, even in ditrysians, the W is not always involved in sex determination[33] and has been lost in several lineages (Saturnidae[34]; see ref.[10] for a review). In the present study, our genomic approach to detect W-derived scaffolds led to the discovery of several female-specific sequences in the WZ *C. ohridella* species. In contrast, no such female-specific sequences were found in *N. degeerella*. Although Lepidoptera W chromosomes consist mainly of ubiquitous repeats[15, 35, 36], which may be difficult to detect in genome assemblies, these results provide complementary evidence for the lack of a W chromosome in Adelidae and support the view that the W was secondarily acquired in either the ancestor of Ditrysia and Tischerioidea[11–14] or early in Ditrysia evolution[15].

What could then be the origin of the ditrysian W chromosome? The favoured alternative hypothesis has been that a B

chromosome may have been recruited to perform female-specific functions, thereby creating a new W chromosome[11]. B chromosomes are supernumerary chromosomes that are found in many insect species (including several Lepidoptera[37, 38]) and are generally thought to be primarily parasitic. Because of their biological similarity to Y/W chromosomes (they are often highly repetitive and gene poor) and ability in some species to segregate with the X at meiosis, they have been suggested as the source of sex-specific chromosomes in other insects[39], including *Drosophila*[4]; but in this case no representative X0 species is available for comparison. Another possibility is that a sequence derived from a symbiont or parasite was domesticated for the purpose of sex determination. Many insects carry endosymbionts, such as *Wolbachia*, which can manipulate the sex ratio of the progeny of infected individuals by interfering with the sex determination pathway (feminizing the progeny gives them an advantage, as they are only transmitted maternally[40]). The horizontal transfer of a sex-determining gene has recently been reported in pillbugs, where the acquisition by an autosome of a *Wolbachia*-derived female-determining gene has led to the formation of a neo-W chromosome[40]. Although a W chromosome fully derived from bacterial DNA has yet to be described, it seems plausible in light of those results. Finally, it is possible that duplication of at least part of a chromosome provided the source material for the evolution of the W chromosome, although the initial aneuploidy resulting from such duplication would have been unlikely to remain stable in the population.

While we cannot at this point distinguish between these hypotheses, our results argue against a canonical model of W chromosome evolution, in which a pair of autosomes is recruited to perform sex determination. Studies of other X0-XY and Z0-WZ pairs of species will shed light on how common such non-canonical origin of sex-specific chromosomes is, and on which of the different hypotheses play a role.

## Methods

**Biological material and sequencing**. Male and female individuals of *T. sylvina* and *N. degeerella* were collected on the IST Austria campus between May 2015 and September 2016. *Cameraria ohridella* males and females were collected on horse-chestnut trees in Vienna (Austria) in July and August 2016. All individuals were kept at −80°C until DNA extraction. DNA was extracted from either a single male or female individual (*T. sylvina*, *N. degeerella* and *C. ohridella*), or pooled individuals (*C. ohridella*: one library from 10 males and one library from 10 females, *N. degeerella*: one library from 9 males). We further re-sequenced the single *T. sylvina* female to improve its low genomic coverage. Specimens were lysed using the Tissue Lyser II (QUIAGEN) and DNA was isolated using the DNeasy blood and tissue kit (QUIAGEN). DNA was then sheared by sonication with a Bioruptor instrument (Diagenode). Library preparation and sequencing (Hi-seq 2500 Illumina, 125 bp paired-end reads) were performed at the Vienna biocenter next generation sequencing facility (Austria). Library information and statistics are provided in Supplementary Table 1.

**Publicly available data**. We downloaded one additional genome sequencing library derived from a single adult[41] for *T. sylvina* (accession number: SRR1190481) and several libraries derived from a pool of females for the Trichoptera species *Limnephilus lunatus* (accession numbers: SRR947083 –SRR947088) from https://www.ncbi.nlm.nih.gov/sra/.

The genome assembly of *L. lunatus* (GCA_000648945.1_Llun_1.0_genomic.fna) was obtained from https://www.ncbi.nlm.nih.gov/assembly.

**De novo genome assemblies**. Sequenced reads in BAM format were first converted into Fastq using Bedtools[42] v2.25.0 and their quality assessed with FastQC[43] v0.11.2. No further quality filter neither trimming was deemed necessary, as the average Phred score was >25 throughout the reads of each library. The genome of each species was assembled from a single paired-end genome sequencing library, as this yielded a more continuous assembly than pooling libraries for each species. Specifically, each library was first individually assembled using SOAPdenovo2[44] with default parameters, using a range of kmer lengths from 33 to 93. The least fragmented assembly, as measured by N50 and number of scaffolds, was chosen for further processing (*C. ohridella*: kmer = 83, library #44577; *N. degeerella*: kmer = 63, library #37850; *T. sylvina*: kmer = 63, library #44580). The chosen assembly was

then improved using GapCloser v1.12 (http://soap.genomics.org.cn/index.html) with default parameters. Scaffolds with a minimum length of 200 bp were selected and reassembled with all libraries (except the re-sequenced *T. sylvina* female #44580 and the 9 *N. degeerella* pooled males #52195) in a hierarchical manner using SSPACE[45] v3.0 with no contig extension option and a minimum of 48 overlapping bases. Finally, *N. degeerella* and *C. ohridella* assemblies were improved using Cap3[46] v02.10.15 with default parameters. The *T. sylvina* assembly was too large for this last step to run without exceeding the memory allowance of our cluster (512 GB RAM), so it was not performed in this species. Genome assembly statistics are provided in Supplementary Table 2 and codes in Supplementary Data 2.

**Read mapping and estimation of genomic coverage**. For each species, all male and female DNA reads were mapped separately to the de novo assembled genomic scaffolds using Bowtie2[47] v2.2.9 with default parameters (--end-to-end --sensitive mode). The resulting alignments were filtered to keep only uniquely mapped reads by selecting lines that did not match "XS:I", as this is the Bowtie2 tag for the score of the second best alignment of the read, and is not present for reads that only have one match. The male and female coverages were then estimated from the filtered SAM files with SOAPcoverage v2.7.7 (http://soap.genomics.org.cn/index.html). The coverage values for each library are provided in Supplementary Data 3–6 and codes in Supplementary Data 7–8.

**Orthology and assignment to the *B. mori* chromosomes**. The gene set of *B. mori* (Supplementary Data 9), as well as the respective gene chromosomal locations (Supplementary Data 10), were obtained from the Silkworm Genome Database (v2.0, http://silkworm.genomics.org.cn). For each sequenced species, the *B. mori* gene set was mapped to the genome assembly using Blat[48] with a translated query and data set, and a minimum mapping score of 50; only the location with the best score was kept for each gene. When several genes overlapped on the same scaffold by more than 20 base pairs, only the gene that had the highest mapping score was kept for that genomic location. Each scaffold was then assigned to the chromosome on which the majority of *B. mori* genes that mapped to it were located. When a scaffold carried the same number of genes from two or more chromosomes, the mapping scores of all genes from each were added, and the scaffold was assigned to the chromosome that had the largest sum of mapping scores. The final chromosomal assignments are provided in Supplementary Data 11–14 and codes in Supplementary Data 15.

**Scaffold classification by the lenient method**. For each species, the per-scaffold coverage was summed-up across samples grouped by sex. Only scaffolds longer than 1500 bp with coverage above 5 and below the 99.5th percentile in the grouped male sample were considered for further analysis. The resulting Log2 of the female over male coverage (Log2(F/M)) distribution was expected to be bimodal. The highest peak corresponds to the autosomal mode, which we call "A_mode" and for which males and females are expected to show equal coverage. The second highest peak corresponds to the Z-linked mode. It should be centered around "A_mode–1", because Z-linked scaffolds are expected to show a two-fold coverage reduction in female relative to male. We classified all scaffolds having a Log2(F/M) value less than "A_mode–0.6" as Z-linked, and more than "A_mode–0.4" as autosomal, based on a *n*-class histogram (*n* = 200 for *C. ohridella*; *n* = 120 for *N. degeerella* and *T. sylvina*). Scaffolds falling in-between were tagged as "unclassified". The scaffold classification is provided in Supplementary Data 1 and the R code used is provided in Supplementary Data 16.

**Scaffold classification by the stringent method**. We took advantage of the homology of the the Z chromosomes of the sequenced species and that of *B. mori* to empirically determine stricter coverage cutoffs for the Z-linked classification. We reasoned that, as the cutoff chosen for Z-assignment becomes more stringent, the percentage of scaffolds classified as Z-linked that are on the Z chromosome of *B. mori* (Chr. 1) should increase, but plateau once (almost) only true Z-linked scaffolds are being selected (this final percentage should roughly correspond to the real proportion of conserved genes between the two Z chromosomes; Supplementary Fig. 2). Under this stringent classification, scaffolds were classified as Z-linked if: (1) they fell below the Log2(F/M) value at which the proportion of the resulting Z-classified scaffolds mapping to *B. mori* Z chromosome has reached a plateau (*C. ohridella*: −0.68; *N. degeerella*: −1.71; *T. sylvina*: −0.08); (2) they had a female coverage below the female median (as the Z chromosome is only present in one copy in females).

Scaffolds were classified as autosomal if they fell above the Log2(F/M) corresponding to the 10th percentile of the Log2 of the female over male coverage of all scaffolds mapping to *B. mori* autosomes. Scaffolds falling between the Z-linked and autosomal limits were tagged as "unclassified". The stringent classification is provided in Supplementary Data 1 and the R code used is provided in Supplementary Data 16.

**Detection of gene movement onto and off the Z chromosome**. We combined the Z/autosome classification from the four Lepidoptera species to determine which genes were ancestrally present on the Z chromosome, and to detect gene movement

along the phylogeny. In the lenient classification, the 69 genes that were deemed to be on the ancestral Lepidoptera Z were Z-linked at least in *T. sylvina* and another lineage. These ancestrally Z-linked genes, plus the eight that were Z-linked in *N. degeerella* and in the two Ditrysia species were classified as ancestral to Ditrysia + *N. degeerella*. Finally, genes that were Z-linked only in the two Ditrysia were classified as having moved onto the Z at the root of the Ditrysia, and genes that were Z-linked in only one species (excluding *T. sylvina*) as having moved in the species-specific lineages. Similar arguments apply for movement off the Z, replacing "Z-linked" with autosomal. We applied this methodology to both the lenient and stringent classifications, and we further tested its robustness by using an extended data set with a minimal cutoff length of 500 bp instead of 1500 bp.

**In silico identification of W-candidates and PCR validation**. W-derived sequences are expected to show no or very low coverage in male samples, and reduced coverage in female samples relative to the autosomal scaffolds (as the W chromosome is only present in one copy in females). Scaffolds larger than 1000 bp that had coverage <1.1 in each male sample, and above ¼ of the median but below the median in each female sample, were therefore designated as candidate W-derived sequences (42 W-candidates in *C. ohridella*, two in *N. degeerella*, and 2114 in *T. sylvina*). Since many of the *T. sylvina* candidates are likely to result from the lower genomic coverage and fragmented genome assembly, and from the fact that we used a single female, we applied more stringent filters (a minimal scaffold size of 2000 bp, and null coverage in each male sample) to select the most promising 23 candidates for PCR testing. Fasta sequences of all tested candidates are provided in Supplementary Data 17–19 and the R code used to identify W-candidates is provided in Supplementary Data 20.

In order to test if these candidates corresponded to female-specific sequences, primers were designed for the two *N. degeerella* candidates, 23 *T. sylvina* candidates, and 10 of the *C. ohridella* candidates, as well as for two autosomal control genes (BGIBMGA013945-TA actin and BGIBMGA009132-TA beta-tubulin *B. mori* orthologs) using Primer 3 plus[49] with default parameters. In *N. degeerella*, we further designed primers for 26 of 27 additional scaffolds that fit the coverage parameters for W-candidates but were below 1000 bp (no primers could be designed for the last one, because its low GC content did not fulfill the optimal requirements of Primer 3 plus). Primer specificity was assessed by independently mapping each primer against its respective genome assembly using Blat (minimum sequence identity of 100%, tile size of 16, step size of 1, and minimum match of 1), and, when possible, non-repetitive primers were picked (some of the shorter *N. degeerella* scaffolds only yielded non-specific primers). A pair of primer was considered specific if the forward and the reverse primers mapped only once to the target scaffold and never simultaneously to other scaffolds. Primer sequences and expected length of PCR products are provided in Supplementary Table 7.

For each species, DNA was extracted from three new males and three new females using the DNeasy blood and tissue kit (QIAGEN). The female-specificity of the sequences was tested in the six individuals using the GoTaq G2 polymerase PCR kit (Promega, Cat#M7841) with an annealing temperature of 52 °C, an elongation time of 1 min, for 35 cycles. Finally, the amplification and size of each product was assessed by electrophoresis on a 1.5% agarose gel, stained with SYBRsafe (Thermo Fisher Scientific).

**Sex identification of the available *T. sylvina* individual**. The published DNA reads were mapped to our de novo *T. sylvina* genome assembly and coverage was estimated following the same procedure as for the other samples. The resulting coverage values were then compared between genes classified as Z-linked and genes classified as autosomal in *T. sylvina*. This initial classification (provided in Supplementary Data 21) was made by only using data from the single male and the single female we collected. Following the lenient method, we applied an upper Z-linked cutoff value of "A_mode−0.6" and a lower autosomal cutoff value of "A_mode−0.4". The median coverage value of Z-linked scaffolds was not significantly different from that of autosomal scaffolds (median_Z = 4.6; median_A = 4.6; $P = 0.383$ with a one-tailed Wilcoxon rank-sum test), as expected for a male having two copies of the Z chromosome. Therefore, we treated the extra individual as a second *T. sylvina* male in our analyses. The R code used is provided in Supplementary Data 22.

**Code availability**. The full bioinformatic pipeline used in this work and inputs are provided as Supplementary Data files.

**Data availability**. All reads generated for this study have been deposited on the NCBI Sequence Read Archive (SRA) under BioProject number PRJNA388200. The de novo genome assemblies generated for this study have been deposited on Lepbase[50] (http://ensembl.lepbase.org).

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

## Acknowledgements

The authors thank Harald Krenn for advice on microlepidoptera, Ariana Macon for DNA extractions and preparation for sequencing, the Vicoso lab, Dominik Schrempf and Chay Graham for taking part in moth collections and James R. Walters, Petr Nguyen, and Anna Voleníková for their comments on the manuscript. This project was funded by an Austrian Science Foundation FWF grant (Project P 28842) to B.V.

## Author contributions

C.F. and M.A.L.P. contributed equally to this work. Experiments, data analyses, writing and conceptualization were performed by C.F., M.A.L.P. and B.V.; funding was acquired by B.V.

## Additional information

**Competing interests:** The authors declare no competing financial interests.

