## [Peer Review File · Nature Communications]

Reviewers' comments:

Reviewer #1 (Remarks to the Author):

Lepidoptera are among the largest groups of animals and certainly the largest group with female heterogamety, characterized by predominating WZ/ZZ sex chromosome system. In this study, authors addressed one of the fundamental questions of Lepidoptera genetics, the origin of the W chromosome, which is an evolutionary novelty in this insect order, using comparative genomics. They sequenced and assembled genomes of females and males of three basal lepidopteran species and used available genomic sequences of the model species, the silkworm *Bombyx mori*, and of a caddisfly species, *Limnephilus lunatus*, a representative of the sister order Trichoptera, which served as an outgroup. Then in the species studied they identified orthologs of *B. mori* Z-linked and autosomal genes using bioinformatic tools and tested two previously published hypothesis and one general hypothesis on the evolution of the lepidopteran sex chromosomes and the origin of the W chromosome. Their comparative genomic analyses brought two important conclusions well reflecting the above mentioned fundamental question: (i) the Z chromosome is highly conserved not only within Lepidoptera but most probably also in their sister group, the Trichoptera; (ii) the W chromosome had not arisen from an autosome, whose homologue fused with the ancestral Z chromosome but results suggest its non-canonical origin. In addition, results of this study suggest a neo-Z chromosome, originating by fusion of the ancestral Z chromosome with an autosome corresponding chromosome 18 in the silkworm, in a representative of lower Ditrysia, *Cameraria ohridella* (Gracillariidae).

The species sequenced were very well selected, one representing lower Ditrysia and two basal non-Ditrysia. Bioinformatic analyses also seem to be well done and their results well support the main conclusions made. I would only suggest more caution in the interpretation of results on the absence of the W chromosome in the representative of Hepialidae, *Triodia sylvina*, in particular because three other species of this family have the W chromosome (see my specific comment below). Although results of PCR tests of putative W-linked scaffolds are negative, this question is in my eyes inconclusive. In a number of lepidopteran species including the model *B. mori*, the W chromosome is mainly composed of ubiquitous repeats, i.e. repeats present in autosomes and the Z chromosome and "only" accumulated in the W chromosome (see several published studies on CGH analysis of the W chromosome composition in *B. mori*, *Cydia pomonella*, and other species and uniformity of W-DNA found by BAC-FISH in *B. mori* and *Biston betularia*). It might be then difficult to impossible to identify female-specific sequences using approaches used in the present study.

Overall, this work is scientifically sound and bring new results of general interest that greatly contribute to understanding fundamental mechanisms of sex chromosome evolution. This work certainly deserves publishing. However, some parts of the manuscript, especially in Introduction and Discussion, require revision.

Specific comments

(1) Line 56 (Introduction): it cannot be said "which shares a homologous pair of ZW sex

chromosomes" because the lepidopteran W and Z chromosomes lack any obvious homology". You may change it to "a heterologous pair of WZ sex chromosomes".

(2) Line 56 (Introduction): from the cited work of Nishikawa et al. (2015), it cannot be deduced that Papilionidae, represented by *Papilio polytes* and *P. Xuthus*, have a large-scale chromosomal Z-autosomal rearrangements. This work reports an autosomal inversion carrying the doublesex gene. This reference and related text (Papilionidae) should be omitted. Instead, you may refer, for example, either Yoshido et al. (2011, DOI: 10.1038/hdy.2010.94) or Yoshido et al. (2016, already cited in your manuscript), which show a Z-chromosome-autosome fusion in wild silkmoths, *Samia cynthia* ssp. (*Saturniidae*).

(3) Lines 91-92: statement that "two outgroup lineages are presumed to be lacking a W due to the absence of sex chromatin" needs references, Traut and Marec (1996) for *Triodia sylvina* (*Hepialidae*) and Lukhtanov (2000) for both *Adelidae* and *Hepialidae*.

(4) Supplementary Text 2a and 2b are missing in the shared Dropbox folder, available on Fraise_Picard_Vicoso_NatComm.SuppInfo

(5) Figure 4: the sex chromosome constitution in *B. mori* is wrong in both A) and B) figures. Replace Z0 with WZ.

(6) Lines 205-06: since it is impossible to identify orthologs of ALL Z-linked genes known in *B. mori*, you should emphasize here that your conclusion is based on TESTED genes.

(7) Lines 222-23: statement that "absence of a W has been confirmed using cytogenetics in the

sister order Trichoptera and in Micropterigidae" needs references

- for Trichoptera:

Marec F, Novák K. 1998. Absence of sex chromatin corresponds with a sex-chromosome univalent in females of Trichoptera. *Eur J Entomol* 95: 197-209.

- for Micropterigidae:

Traut W, Marec F. 1997. Sex chromosome differentiation in some species of Lepidoptera (*Insecta*). *Chromosome Res* 5: 283-291.

(8) Lines 228-34: I agree with the conclusion that *Adelidae* lack a W chromosome, which seems to be well supported by negative results of a relatively high number of PCR-tested sequences from putative W-linked scaffolds in *N. degeerella*. However, I cannot agree with the statement that "These results provide complementary evidence that a W is lacking in these two basal lineages, and suggest that the W was independently acquired in some *Hepialidae* lineages", because based on available data the W is lacking only in one lineage (*Adelidae*), while in *Hepialidae* it is evidently present in three different genera (*Endoclita*, *Phymatopus*, and *Hepialus*) and its absence in *Triodia sylvina* is questionable due to only a small subset tested and also due to a lower quality of the assembly as the authors mentioned in Results. Even if the W really is missing in *T. sylvina*, it is more parsimonious to interpret this finding as loss of the W within *Hepialidae* than the independent origin of the W

within Hepialidae. In addition, *Phymatopus*, *Hepialus*, and *Triodia* are probably closely related genera as they were earlier classified all as the genus *Hepialus*.

(9) Line 238: statement that "B-chromosomes are found in many insect species (including several Lepidoptera)" needs to be supported with references, especially B chromosomes in Lepidoptera, which are important for the hypothesis on a non-canonical origin of the W chromosome.

Minor suggestions

Throughout the manuscript: omit hyphen between Z or W or B and chromosome

Throughout the manuscript: ZW to WZ (explanation: it is custom to order these symbols alphabetically, see main reviews on sex chromosomes of Lepidoptera)

Throughout the manuscript and Supplementary Information: use only "bp" for base pairs instead of "bps"

Throughout the manuscript and Supplementary Tables: "P" as probability should be italicized. Be consistent (in Suppl. Tables 6 and 7, both capital "P" and small "p" is used), preferably use capital "P".

Throughout the manuscript: adjectives for Ditrysia and non-Ditrysia, i.e. ditryisian(s) and non-ditryisian(s), should be written with small initial "d"

Line 25: Moths and butterflies (Lepidoptera)

Line 80: Homoptera: Kuznetsova et al. 1997

Line 89: "horse chestnut moth" is usually named "horse-chestnut leaf miner" (see: <http://www.leps.it/>) or "horse chestnut leaf-miner" (see: <https://www.ukmoths.org.uk/species/cameraria-ohridella>) "horsechestnut leafminer" (see <http://www.cabi.org/isc/datasheet/40598>)

Line 90: instead of "a fairy moth", better to use either "a fairy longhorn moth" or "the yellow-barred long-horn" (see <https://www.norfolkmoths.co.uk/micros.php?bf=1480>)

Line 225: a WZ1Z2 karyotype [1 and 2 should be written as subscripts]

Line 433: I think that the correct journal title is "Phegea"; whereas "Vlaamse Vereniging Voor Entomologie" is the name of "Flemish Entomological Society"

Line 445: microlepidopterous

Lines 448-9: Kuznetsova VG, Nokkala S, Maryńska-Nadachowska A. 1997. Karyotypes, sex chromosome systems, and male meiosis in Finnish psyllids (Homoptera: Psylloidea). *Folia Biol* 45: 143–152.

Lines 477-8: use small initials in all words of the title of Pal and Vicoso (2015) citation, except the first word and X

Line 492: Traut W. 1999.

Lines 505-07: Citation of Voleníková (2015) should be probably cited as follows:
Voleníková A. 2015. Karyotype and sex chromosomes analysis of two species from basal lepidopteran family Hepialidae (Lepidoptera: Hepialidae). Master thesis (in Czech), University of South Bohemia, České Budějovice, Czech Republic.

Frantisek Marec

3 June 2017

Reviewer #2 (Remarks to the Author):

This manuscript presents the sequencing of multiple moth genomes in order to uncover the origin of the Lepidoptera Z chromosome, and as a consequence the W chromosome. The origin of the W chromosome has been an evolutionary mystery as it doesn't seem to follow the canonical model for the origin of sex chromosomes.

I really enjoyed this manuscript. The analyses are well done and generally clear, and the results are exciting. I think a lot of people will be interested in the results, and that this will spur similar work in other systems, as well as work on the W chromosome that I will be looking forward to.

All that said, I have a few comments about the writing and clarity of presentation, though none of these concerns is major.

-The phylogenetic relationships among all the lineages and species discussed here were very confusing. The paper definitely needs a phylogenetic tree with clear labels as one of the first figures presented--the current Figure 4 does not show all of the different groups that are discussed. The manuscript uses terms like Micropterigidae, Trichoptera, Ditrysia, but I think only the last one is labeled in Figure 4. Is Trichoptera shown anywhere?

In addition, it was hard to understand some of the language being used about these lineages. The three species sequenced are discussed as "basal" Lepidoptera, with two of them further referred to as "outgroups." Both of these terms are hard to understand in context (outgroups are used for rooting trees), and basal should never be used to refer to extant lineages (especially "most basal"). A better phrase than basal should be found, or the species should simply be referred to as "non-Ditrysia" (or similar).

-I really appreciated all of the follow-up work that went into trying to determine whether a W chromosome exists in *T. sylvina* and *N. degeerella*. I think it is clear that *N. degeerella* does not have a W. But it was unclear exactly what work was done to check *T. sylvina*. The text says "an exhaustive examination" was unfeasible, but it was not clear to me what the results of any PCRs was: the text also says of the candidate W genes that "only a small subset was tested". This should all be made clearer.

-line 56: it was hard to understand "homologous pair of ZW sex chromosomes" in this context, since as far as I understand there is no W in *B. mori*, and the W may have been gained multiple times.

-line 66: there are other insects with female heterogamety, so why the need to qualify with "only major"?

-line 152: is "DNA-seq" a commonly used phrase?

-line 154: I don't think a p-value can ever truly be "0". Maybe change to " $p < 0.001$ " or whatever is most appropriate.

-line 200 (approximately): I think it would be useful to the reader to state that the Lepidoptera Z is not homologous to the X chromosome in the dipterans *D. melanogaster* and *A. gambiae*, and presumably the ancestral dipteran either. This would help to put the time of the existence of the Z into context. An appropriate citation is Pease and Hahn (2012, MBE) or possibly Vicoso and Bachtrog (2015).

-line 202: remove the "to" at the beginning of the line.

Reviewer #3 (Remarks to the Author):

The authors took a comparative genomics approach to explore how female heterogametic sex chromosomes have evolved in Lepidoptera. The authors find support for a single origin of the Z chromosome in the common ancestor of Lepidoptera and Trichoptera. The absence of a W chromosome from newly sequenced basal taxa suggests the W may have been independently derived in a number of taxa. Overall I found the analyses thorough and I do not have any major comments on the experiments. I have some minor comments on aspects of the manuscript that I did not find completely clear.

Line 75: In figure 1, the authors graphically illustrate different models for secondary acquisition of the W chromosome. The color schemes, sizes, and notations of the chromosomes make it a little confusing to follow which autosomes the W chromosome derived from. For instance in the Z-autosome fusion model, the W is a derivative of A1. It would be clearer if the W chromosome was illustrated as the same size as A1 or if the authors adopted a different notation that indicates the origin of the W (ex. W-A1 and Z-A1). For the sex-chromosome turnover model, the previous Z and W chromosomes are lost and become autosomes. The authors name these chromosomes A1, but these are not homologous to the A1 chromosome in the ancestral female karyotype. These should have a unique autosome name, like A3.

Line 122: The authors mapped annotated *B. mori* genes to the scaffolds from the de novo assembly. Many of these scaffolds are very short, especially in *T. sylvina* with an N50 of 1,184 bp. Did the authors require that a complete gene annotation aligned to a given scaffold, or did they also allow for portions of genes to align to smaller scaffolds? Allowing for partial gene alignments would increase the number of genes available for their analyses if this were not done.

In addition, rather than limiting the analysis to genes that mapped to de novo assembled scaffolds, the authors could align the sequencing reads to the *B. mori* gene set as a reference genome. This could increase the overall gene count available for the analyses.

Lines 147-151. The authors found that mapped Z-linked genes were distributed throughout

chromosome 1, rather than being restricted to certain regions of the chromosomes. The authors explain that clustering of the genes would be expected if there was a Z-autosome fusion, but this reasoning is not clear to me. If there was a fusion, I would expect a pattern like that observed in *C. ohridella*, where a second autosome displays Z-specific coverage patterns. Clustering of genes to certain regions of chromosome 1 would occur if there was a lineage-specific deletion of the Z-chromosome. The authors need to explain their reasoning more clearly how this would be evidence for a fusion.

Lines 213-234. The authors discovered additional species that do not have a W chromosome. Combined with previous studies, the authors conclude that this is evidence the W chromosome was independently derived in other Lepidopteron lineages. Based on the descriptions in the text, it is hard to follow if this is the most parsimonious explanation. I think a figure showing a phylogeny combining all the taxa where the presence/absence of the W chromosome has been investigated would be helpful. With this trait mapped onto the tree, it would be easy to see if there are more gains or losses.

Figure 3. There appears to be a large clustering of scaffolds in *C. ohridella* that follow an autosomal coverage pattern (blue dots). Do these represent scaffolds that have moved off of the Z chromosome? If so, why do the relative numbers of blue scaffolds in Figure 3 not match up with the gene counts in Figure 4? For each species in Figure 4, there are only a handful of genes that have moved off of the Z chromosome.

We would like to thank the reviewers for their helpful feedback, and enthusiasm for the manuscript (now entitled "The deep conservation of the Lepidoptera Z chromosome suggests a non-canonical origin of the W" to fulfil the editorial requirements of Nature Communications).

We have addressed each of the comments below, and have also added two more datasets in order to strengthen our conclusions:

1. Mirroring the analysis performed on *C. ohridella*, we sequenced a pool of 9 *N. degeerella* males. This greatly increased our power to detect candidate W-sequences in this species, so that we could now exclude female-specificity of all but one candidate W-sequences, independently of their size (whereas before only scaffolds larger than 1,000 bp were tested; primers could not be designed for a single scaffold). Our assignment to the Z and autosomes based on coverage was also improved by using this extra data.

2. To increase the coverage of the *T. sylvina* genomic scaffolds, we re-sequenced the female library at a greater depth.

Importantly, none of our previous conclusions were affected by adding these two datasets.

All the new reads have been deposited on the NCBI Sequence Read Archive (SRA) under BioProject number PRJNA388200: they will be released upon publication. The three *de novo* genome assemblies generated for this study will be deposited on Lepbase (lepbase.org): the url access will be provided in the final manuscript.

Detailed answers to the reviewers' comments:

Reviewer #1 (Remarks to the Author):

Lepidoptera are among the largest groups of animals and certainly the largest group with female heterogamety, characterized by predominating WZ/ZZ sex chromosome system. In this study, authors addressed one of the fundamental questions of Lepidoptera genetics, the origin of the W chromosome, which is an evolutionary novelty in this insect order, using comparative genomics. They sequenced and assembled genomes of females and males of three basal lepidopteran species and used available genomic sequences of the model species, the silkworm *Bombyx mori*, and of a caddisfly species, *Limnephilus lunatus*, a representative of the sister order Trichoptera, which served as an outgroup. Then in the species studied they identified orthologs of *B. mori* Z-linked and autosomal genes using bioinformatic tools and tested two previously published hypothesis and one general hypothesis on the evolution of the lepidopteran sex chromosomes and the origin of the W chromosome. Their comparative genomic analyses brought two important conclusions well reflecting the above mentioned fundamental question: (i) the Z chromosome is highly conserved not only within Lepidoptera but most probably also in their sister group, the Trichoptera; (ii) the W chromosome had not arisen from an autosome, whose homologue fused with the ancestral Z chromosome but results suggest its non-canonical origin. In addition, results of this study suggest a neo-Z chromosome, originating by fusion of the ancestral Z chromosome with an autosome

corresponding chromosome 18 in the silkworm, in a representative of lower Ditrysia, *Cameraria ohridella* (Gracillariidae).

The species sequenced were very well selected, one representing lower Ditrysia and two basal non-Ditrysia. Bioinformatic analyses also seem to be well done and their results well support the main conclusions made. I would only suggest more caution in the interpretation of results on the absence of the W chromosome in the representative of Hepialidae, *Triodia sylvina*, in particular because three other species of this family have the W chromosome (see my specific comment below). Although results of PCR tests of putative W-linked scaffolds are negative, this question is in my eyes inconclusive. In a number of lepidopteran species including the model *B. mori*, the W chromosome is mainly composed of ubiquitous repeats, i.e. repeats present in autosomes and the Z chromosome and “only” accumulated in the W chromosome (see several published studies on CGH analysis of the W chromosome composition in *B. mori*, *Cydia pomonella*, and other species and uniformity of W-DNA found by BAC-FISH in *B. mori* and *Biston betularia*). It might be then difficult to impossible to identify female-specific sequences using approaches used in the present study.

Overall, this work is scientifically sound and bring new results of general interest that greatly contribute to understanding fundamental mechanisms of sex chromosome evolution. This work certainly deserves publishing. However, some parts of the manuscript, especially in Introduction and Discussion, require revision.

We thank the reviewer for these very helpful comments, which we address individually below:

Specific comments

1.1

Line 56 (Introduction): it cannot be said “which shares a homologous pair of ZW sex chromosomes” because the lepidopteran W and Z chromosomes lack any obvious homology”. You may change it to “a heterologous pair of WZ sex chromosomes”.

This was changed as suggested (1.64)

1.2

Line 56 (Introduction): form the cited work of Nishikawa et al. (2015), it cannot be deduced that Papilionidae, represented by *Papilio polytes* and *P. Xuthus*, have a large-scale chromosomal Z-autosomal rearrangements. This work reports an autosomal inversion carrying the doublesex gene. This reference and related text (Papilionidae) should be omitted. Instead, you may refer, for example, either Yoshido et al. (2011, DOI: 10.1038/hdy.2010.94) or Yoshido et al. (2016, already cited in your manuscript), which show a Z-chromosome-autosome fusion in wild silkmths, *Samia cynthia* ssp. (Saturniidae).

We thank the reviewer for the correction. We removed the reference to Papilionidae, and referred to the Z-autosome fusion in Saturniidae instead, by citing “Yoshido et al. 2011” (1.67)

1.3

Lines 91-92: statement that “two outgroup lineages are presumed to be lacking a W due to the absence of sex chromatin” needs references, Traut and Marec (1996) for *Triodia sylvina* (Hepialidae) and Lukhtanov (2000) for both Adelidae and Hepialidae.

As suggested by the reviewer, we now refer to “Traut and Marec (1996)” and “Lukhtanov (2000)” (1.80-82)

1.4

Supplementary Text 2a and 2b are missing in the shared Dropbox folder, available on Fraise_Picard_Vicoso_NatComm.SuppInfo

We fixed this oversight, and carefully checked that all supplementary files are present. The updated Dropbox folder is now:

<https://www.dropbox.com/sh/c3rvzlkq3plxwsl/AABcTWXIE-gnSNcG6xNobHNja?dl=0>

The Supplementary Text 2a and 2b correspond now to the Supplementary Method 2 (page 6 to 9 of the Supplementary Information word document)

1.5

Figure 4: the sex chromosome constitution in *B. mori* is wrong in both A) and B) figures. Replace Z0 with WZ.

We fixed this error.

1.6

Lines 205-06: since it is impossible to identify orthologs of ALL Z-linked genes known in *B. mori*, you should emphasize here that your conclusion is based on TESTED genes.

We now specify (1.223-226) that results are based on the subset of genes that were classified: “The gene content of the Z chromosome has also remained remarkably stable: 67% of the 138 *T. sylvina* genes that were assigned to the Z-chromosome are also Z-linked in *B. mori*, and this number increases to 93% for *N. degeerella* (out of 300 Z-linked genes) (Supplementary Table 5).”

We have also made it clear in the text which genes were used for our analysis of gene movement: “Fig. 4A, which is based on the 4,132 orthologs that were assigned to the Z or autosomes in all species using the lenient classification; Fig. 4B, which is based on the 3,189 genes classified using the stringent classification” (1.173-175).

Finally, to clarify the gene sample size of each pairwise comparison with *B. mori*, we now report the detailed gene counts in the new Supplementary Table 5.

1.7

Lines 222-23: statement that “absence of a W has been confirmed using cytogenetics in the sister order Trichoptera and in Micropterigidae” needs references

- for Trichoptera:

Marec F, Novák K. 1998. Absence of sex chromatin corresponds with a sex-chromosome univalent in females of Trichoptera. *Eur J Entomol* 95: 197-209.

- for Micropterigidae:

Traut W, Marec F. 1997. Sex chromosome differentiation in some species of Lepidoptera (Insecta). *Chromosome Res* 5: 283-291.

Thank you for suggesting these references. We now referred to “Marec and Nock (1998)” and “Traut and Marec (1997)” (1.243-245).

1.8

Lines 228-34: I agree with the conclusion that Adelidae lack a W chromosome, which seems to be well supported by negative results of a relatively high number of PCR-tested sequences from putative W-linked scaffolds in *N. degeerella*. However, I cannot agree with the statement that “These results provide complementary evidence that a W is lacking in these two basal lineages, and suggest that the W was independently acquired in some Hepialidae lineages”, because based on available data the W is lacking only in one lineage (Adelidae), while in Hepialidae it is evidently present in three different genera (*Endoclita*, *Phymatopus*, and *Hepialus*) and its absence in *Triodia sylvina* is questionable due to only a small subset tested and also due to a lower quality of the assembly as the authors mentioned in Results. Even if the W really is missing in *T. sylvina*, it is more parsimonious to interpret this finding as loss of the W within Hepialidae than the independent origin of the W within Hepialidae. In addition, *Phymatopus*, *Hepialus*, and *Triodia* are probably closely related genera as they were earlier classified all as the genus *Hepialus*.

We now refrain from making claims about the lack of the W in *T. sylvina*, as we agree that our data is not conclusive on this matter. For instance, the results (1.211-213) have been changed to: « These results support the absence of a W-chromosome in *N. degeerella*. Unfortunately the poorer quality of the assembly makes such an exhaustive examination unfeasible in *T. sylvina*. »

1.9

Line 238: statement that “B-chromosomes are found in many insect species (including several Lepidoptera)” needs to be supported with references, especially B chromosomes in Lepidoptera, which are important for the hypothesis on a non-canonical origin of the W chromosome.

The following references were added:

37. Bigger, T. R. L. Karyotypes of Three Species of Lepidoptera Including an Investigation of B-chromosomes in *Pieris*. *Cytologia*. 41, 261–282 (1976).

38. Pearse, F. K. & Ehrlich, P. R. B chromosome variation in *Euphydryas colon* (Lepidoptera: Nymphalidae). *Chromosoma* 73, 263–274 (1979).

1.10

Minor suggestions

Throughout the manuscript: omit hyphen between Z or W or B and chromosome
Change made.

Throughout the manuscript: ZW to WZ (explanation: it is custom to order these symbols alphabetically, see main reviews on sex chromosomes of Lepidoptera)
Change made.

Throughout the manuscript and Supplementary Information: use only “bp” for base pairs instead of “bps”
Change made.

Throughout the manuscript and Supplementary Tables: “P” as probability should be italicized. Be consistent (in Suppl. Tables 6 and 7, both capital “P” and small “p” is used), preferably use capital “P”.
Change made.

Throughout the manuscript: adjectives for Ditryisia and non-Ditryisia, i.e. ditryisian(s) and non-ditryisian(s), should be written with small initial “d”
Change made.

Line 25: Moths and butterflies (Lepidoptera)
Change made 1.33.

Line 80: Homoptera: Kuznetsova et al. 1997
Change made 1.88.

Line 89: “horse chestnut moth” is usually named “horse-chestnut leaf miner” (see: <http://www.leps.it/>) or “horse chestnut leaf-miner” (see: <https://www.ukmoths.org.uk/species/cameraria-ohridella>) “horsechestnut leafminer” (see <http://www.cabi.org/isc/datasheet/40598>)
Change made 1.98.

Line 90: instead of “a fairy moth”, better to use either “a fairy longhorn moth” or “the yellow-barred long-horn” (see <https://www.norfolkmoths.co.uk/micros.php?bf=1480>)
Change made 1.99.

Line 225: a WZ1Z2 karyotype [1 and 2 should be written as subscripts]
Change made 1.247.

Line 433: I think that the correct journal title is “Phegea”; whereas “Vlaamse Vereniging Voor Entomologie” is the name of “Flemish Entomological Society”
The bibliography was updated.

Line 445: microlepidopterous
The bibliography was updated.

Lines 448-9: Kuznetsova VG, Nokkala S, Maryańska-Nadachowska A. 1997. Karyotypes, sex chromosome systems, and male meiosis in Finnish psyllids (Homoptera: Psylloidea). *Folia Biol* 45: 143–152.
The bibliography was updated.

Lines 477-8: use small initials in all words of the title of Pal and Vicoso (2015) citation, except the first word and X
The bibliography was updated.

Line 492: Traut W. 1999.
The bibliography was updated.

Lines 505-07: Citation of Voleníková (2015) should be probably cited as follows:
Voleníková A. 2015. Karyotype and sex chromosomes analysis of two species from basal lepidopteran family Hepialidae (Lepidoptera: Hepialidae). Master thesis (in Czech), University of South Bohemia, České Budějovice, Czech Republic.
The bibliography was updated and this suggestion was taken into account.

Reviewer #2 (Remarks to the Author):

This manuscript presents the sequencing of multiple moth genomes in order to uncover the origin of the Lepidoptera Z chromosome, and as a consequence the W chromosome. The origin of the W chromosome has been an evolutionary mystery as it doesn't seem to follow the canonical model for the origin of sex chromosomes.

I really enjoyed this manuscript. The analyses are well done and generally clear, and the results are exciting. I think a lot of people will be interested in the results, and that this will spur similar work in other systems, as well as work on the W chromosome that I will be looking forward to.

All that said, I have a few comments about the writing and clarity of presentation, though none of these concerns is major.

We thank the reviewer for his enthusiasm and comments, which we address individually below:

2.1

The phylogenetic relationships among all the lineages and species discussed here were very confusing. The paper definitely needs a phylogenetic tree with clear labels as one of the first figures presented--the current Figure 4 does not show all of the different groups that are discussed. The manuscript uses terms like Micropterigidae, Trichoptera, Ditrysia, but I think only the last one is labeled in Figure 4. Is Trichoptera shown anywhere?

We have added a phylogenetic tree (new Figure 1A) of species in this study, and some additional ones in which the presence / absence of the W chromosome has been evaluated. We also report on the phylogeny if the state of the W chromosome was assessed based on karyotypes or sex chromatin bodies.

2.2

In addition, it was hard to understand some of the language being used about these lineages. The three species sequenced are discussed as “basal” Lepidoptera, with two of them further referred to as “outgroups.” Both of these terms are hard to understand in context (outgroups are used for rooting trees), and basal should never be used to refer to extant lineages (especially “most basal”). A better phrase than basal should be found, or the species should simply be referred to as “non-Ditrysia” (or similar).

We thank the reviewer for helping us to clarify the phylogenetic vocabulary used. Throughout the manuscript, we replaced “basal” and “outgroup” by “Monotrysia” (when this was correct), or changed the phrasing entirely.

2.3

I really appreciated all of the follow-up work that went into trying to determine whether a W chromosome exists in *T. sylvina* and *N. degeerella*. I think it is clear that *N. degeerella* does not have a W. But it was unclear exactly what work was done to check *T. sylvina*. The text says “an exhaustive examination” was unfeasible, but it was not clear to me what the results of any PCRs was: the text also says of the candidate W genes that “only a small subset was tested”. This should all be made clearer.

We now specify the number of scaffolds that were tested in the other species: " We further tested the female-specificity of all (in *N. degeerella*) or a subset (10 in *C. ohridella* and 23 in *T. sylvina*) of these W-candidates by PCR amplification in an additional three male and three female individuals (Fig. 5)." (l.200-203)

We have also removed any inferences on the presence of a W in *T. sylvina*, as the number of scaffolds that we tested for female-specificity is too small to be conclusive.

2.4

line 56: it was hard to understand “homologous pair of ZW sex chromosomes” in this context, since as far as I understand there is no W in *B. mori*, and the W may have been gained multiple times.

Following the suggestion of Reviewer #1, we now refer to "a heterologous pair of WZ sex chromosomes".

2.5

- line 66: there are other insects with female heterogamety, so why the need to qualify with “only major”?

We removed “only” l.74-75: " Since Lepidoptera and Trichoptera are sister orders and two major insect lineages with female-heterogamety, this system is thought to have originated in their common ancestor."

2.6

-line 152: is “DNA-seq” a commonly used phrase?

We replaced “DNA-seq” by “genome sequencing” (l.163, l.301, l.312).

2.7

-line 154: I don't think a p-value can ever truly be “0”. Maybe change to “ $p < 0.001$ ” or whatever is most appropriate.

We replaced “0” by “ $P < 0.0001$ ” (l.165).

2.8

-line 200 (approximately): I think it would be useful to the reader to state that the Lepidoptera Z is not homologous to the X chromosome in the dipterans *D. melanogaster* and *A. gambiae*, and presumably the ancestral dipteran either. This would help to put the time of the existence of the Z into context. An appropriate citation is Pease and Hahn (2012, MBE).

We now begin our discussion by mentioning that sex chromosomes are not overall conserved in insects, and cite Pease and Hahn (2012) « Previous work has shown that X and Z sex chromosomes have evolved from independent pairs of ancestral chromosomes in winged insects²³. » (l.216-217)

2.9

-line 202: remove the “to” at the beginning of the line.

Change made.

Reviewer #3 (Remarks to the Author):

The authors took a comparative genomics approach to explore how female heterogametic sex chromosomes have evolved in Lepidoptera. The authors find support for a single origin of the Z chromosome in the common ancestor of Lepidoptera and Trichoptera. The absence of a W chromosome from newly sequenced basal taxa suggests the W may have been independently derived in a number of taxa. Overall I found the analyses thorough and I do not have any major comments on the experiments. I have some minor comments on aspects of the manuscript that I did not find completely clear.

We have now implemented the suggestions below, which we agree helped to clarify the manuscript:

3.1

Line 75: In figure 1, the authors graphically illustrate different models for secondary acquisition of the W chromosome. The color schemes, sizes, and notations of the chromosomes make it a little confusing to follow which autosomes the W chromosome derived from. For instance in the Z-autosome fusion model, the W is a derivative of A1. It would be clearer if the W chromosome was illustrated as the same size as A1 or if the authors adopted a different notation that indicates the origin of the W (ex. W-A1 and Z-A1). For the sex-chromosome turnover model, the previous Z and W chromosomes are lost and become

autosomes. The authors name these chromosomes A1, but these are not homologous to the A1 chromosome in the ancestral female karyotype. These should have a unique autosome name, like A3.

We modified the Figure 1B (previously labelled Figure 1) as suggested.

3.2

Line 122: The authors mapped annotated *B. mori* genes to the scaffolds from the de novo assembly. Many of these scaffolds are very short, especially in *T. sylvina* with an N50 of 1,184 bp. Did the authors require that a complete gene annotation aligned to a given scaffold, or did they also allow for portions of genes to align to smaller scaffolds? Allowing for partial gene alignments would increase the number of genes available for their analyses if this were not done.

As indicated 1.341-343, we used the default Blat configuration with a minimum mapping score of 50 to detect the 1:1 orthologs. This allows for partial alignments in both the query and the reference. This information is reported in the output table of Blat in the columns: tStart (tEnd) -- alignment start (end) position of the reference, and qStart (qEnd) -- alignment start (end) position of the query.

3.3

In addition, rather than limiting the analysis to genes that mapped to de novo assembled scaffolds, the authors could align the sequencing reads to the *B. mori* gene set as a reference genome. This could increase the overall gene count available for the analyses.

Given the large divergence between *B. mori* and *T. sylvina* (~150 myr, Misof et al. 2014), direct mapping of *T. sylvina* reads to *B. mori* scaffolds is expected to be limited. For instance, when we map the *T. sylvina* reads of the single female (library 44580) to the *B. mori* cds using the same Bowtie2 settings that we applied in the manuscript, we obtained a very low overall alignment rate of 0.02%.

To overcome the limitation of the poor *T. sylvina* assembly on our ability to assign genes to chromosomes, we additionally applied a less stringent minimal scaffold length of 500 bp instead of 1,500 bp (new Supp. Figure 3 and new Supp. Table 7). In the pairwise comparison with *B. mori*, the number of genes retained in *T. sylvina* increased from 5053 to 7162 over a total of 7545 homologs. Moreover, there was an increase of the total number of genes available for the gene movement analysis from 4132 to 5831 (lenient classification). This did not change our conclusions, and the absence of Z-autosome fusion in the common ancestor of the Ditrysia was still supported.

3.4

Lines 147-151. The authors found that mapped Z-linked genes were distributed throughout chromosome 1, rather than being restricted to certain regions of the chromosomes. The authors explain that clustering of the genes would be expected if there was a Z-autosome fusion, but this reasoning is not clear to me. If there was a fusion, I would expect a pattern

like that observed in *C. ohridella*, where a second autosome displays Z-specific coverage patterns. Clustering of genes to certain regions of chromosome 1 would occur if there was a lineage-specific deletion of the Z-chromosome. The authors need to explain their reasoning more clearly how this would be evidence for a fusion.

If a secondary fusion occurred in Ditrysia (*B. mori* and *C. ohridella* in our study), the derived Z chromosome of *B. mori* would be composed of the ancestral Z, plus a new Z-linked arm. Since we are using *B. mori* chromosomes as our reference, this would be reflected in only the ancestral part of the Z chromosome showing reduced female coverage in non-ditrysians. The reasoning differs in *C. ohridella* is because in this case, we are using the “ancestral” configuration of *B. mori* as our reference. While the reviewer is correct that part of the Z chromosome becoming autosomal in one of the non-ditrysiian lineages would produce the same pattern, the same independent rearrangement in the non-ditrysiian lineages would have to be invoked to account for this in both *T. sylvina* and *N. degeerella*.

3.5

Lines 213-234. The authors discovered additional species that do not have a W chromosome. Combined with previous studies, the authors conclude that this is evidence the W chromosome was independently derived in other Lepidopteran lineages. Based on the descriptions in the text, it is hard to follow if this is the most parsimonious explanation. I think a figure showing a phylogeny combining all the taxa where the presence/absence of the W chromosome has been investigated would be helpful. With this trait mapped onto the tree, it would be easy to see if there are more gains or losses.

We have now added a new figure (1A) that includes the phylogeny of the species used here, as well as a few more that are particularly relevant to the discussion, and the evidence for presence or absence of a W-chromosome in each. We should note that not all monotrysiian lineages that have been found to lack sex chromatin are shown (as we thought it would crowd the figure; a more complete phylogeny can be found in Dalíková *et al.* (Heredity, 2017)); these of course increase the support for a secondary acquisition of the W, and we would be happy to add them if it is felt that this would strengthen the manuscript.

3.6

Figure 3. There appears to be a large clustering of scaffolds in *C. ohridella* that follow an autosomal coverage pattern (blue dots). Do these represent scaffolds that have moved off of the Z chromosome? If so, why do the relative numbers of blue scaffolds in Figure 3 not match up with the gene counts in Figure 4? For each species in Figure 4, there are only a handful of genes that have moved off of the Z chromosome.

The blue genes in *C. ohridella* mapping to the Z chromosome of *B. mori* (Figure 3A) can either correspond to genes ancestrally Z-linked which moved off the Z in *C. ohridella*, or from genes ancestrally autosomal which moved onto the Z in *B. mori*. Actually, the pattern of gene movements cannot be deduced from pairwise comparisons only, so we developed a method based on all four species (Figure 4).

In Figure 2 and Figure 3, we identified for each species independently the orthologs to the *B. mori* genes. The number of Z-linked and autosomal genes identified for each species is reported in Supp. Table 4. Moreover, we added a new Supp. Table 5, which reports the

number of Z-linked and autosomal genes in common between *B. mori* and each of the sequenced species.

In Figure 4 and Supp. Data 6, we analysed the subset of genes that were assigned to a chromosome in *all* four species. As a consequence, the number of genes retained was greatly reduced. We now made this more explicit by reporting the number of genes falling in each category (*Z vs autosomes*) at the top of Figure 4 and Supp. Figure 3.

REVIEWERS' COMMENTS:

Reviewer #1 (Remarks to the Author):

I am very satisfied with the revision. All points raised in my previous review have been perfectly addressed.

I have only one additional comment. In the revision, authors used an older taxonomic term for non-ditrysiian Lepidoptera, the so-called Monotryisia. However, this term was earlier used only for Heteroneura, a taxon, which include Incurvarioidea (e.g. Adelidae) and Nepticuloidea but not Hepialidae and Micropterigidae. Hepialidae are taxonomically classified into Exoporia and Micropterigidae into Zeugloptera (https://en.wikipedia.org/wiki/Taxonomy_of_the_Lepidoptera; also see Regier et al. 2015, doi: 10.1111/syen.12129). Therefore, the terms "Monotryisia" or "monotryisian" are not correctly used in the text of your work. In addition, Monotryisia is not officially recognized taxon. I suggest to replace these terms with non-Ditryisia and non-ditrysiian, respectively. This also regards Fig. 1A – in this case you may either replace Monotryisia above the line with non-Ditryisia or simply delete Monotryisia and also delete the line.

A similar problem is now in the revised legend of Fig. 2, where authors replaced (as suggested by Reviewer 2) "basal Lepidoptera" with a more specific term. However, they replaced it with "distant monotrysiian species", which is completely wrong, because C. ohridella belongs to lower Ditryisia, N. degeerella to Heteroneura and T. sylvina to Exoporia, the latter two represent non-Ditryisia but not Monotryisia. I like the previous term "basal Lepidoptera", which is perfectly correct and fully sufficient for the Fig. 2 legend.

Minor suggestions

Line 189: $n=31$ (the symbol for chromosome number should always be written with small letter)

Lines 487-489: you may add DOI: 10.1093/jhered/esx063 to the citation of Dalíková et al. (2017)

Frantisek Marec
16 August 2017

Reviewer #2 (Remarks to the Author):

The revisions look great, especially the new Figure 1.

Reviewer #3 (Remarks to the Author):

The authors have addressed all of my comments satisfactorily. I have no additional

comments on the revised draft. This work will be of broad interest to the readers of Nature Communications.

Reviewer #1 (Remarks to the Author):

I am very satisfied with the revision. All points raised in my previous review have been perfectly addressed.

I have only one additional comment. In the revision, authors used an older taxonomic term for non-ditrysiian Lepidoptera, the so-called Monotryisia. However, this term was earlier used only for Heteroneura, a taxon, which include Incurvarioidea (e.g. Adelidae) and Nepticuloidea but not Hepialidae and Micropterigidae. Hepialidae are taxonomically classified into Exoporia and Micropterigidae into Zeugloptera (<https://en.wikipedia.org/>; also see Regier et al. 2015, doi: 10.1111/syen.12129). Therefore, the terms “Monotryisia” or “monotryisian” are not correctly used in the text of your work. In addition, Monotryisia is not officially recognized taxon. I suggest to replace these terms with non-Ditryisia and non-ditrysiian, respectively. This also regards Fig. 1A – in this case you may either replace Monotryisia above the line with non-Ditryisia or simply delete Monotryisia and also delete the line.

We thank the reviewer for the correction, and we followed his suggestion. We replaced the terms “Monotryisia” and “monotryisian” by “non-Dytrisia” and “non-dytrisian” throughout the manuscript, and in Figure 1.

A similar problem is now in the revised legend of Fig. 2, where authors replaced (as suggested by Reviewer 2) “basal Lepidoptera” with a more specific term. However, they replaced it with “distant monotryisian species”, which is completely wrong, because *C. ohridella* belongs to lower Ditryisia, *N. degeerella* to Heteroneura and *T. sylvina* to Exoporia, the latter two represent non-Ditryisia but not Monotryisia. I like the previous term “basal Lepidoptera”, which is perfectly correct and fully sufficient for the Fig. 2 legend.

We thank again the reviewer to point out the error, and revised the legend of Figure 2. We changed “The Z chromosome of *B. mori* (Chr. 1) is homologous to the Z chromosome of distant monotryisian species” to “The Z chromosome of *B. mori* is homologous to that of the other species”.

Minor suggestions

Line 189: n=31 (the symbol for chromosome number should always be written with small letter)

Change made.

Lines 487-489: you may add DOI: 10.1093/jhered/esx063 to the citation of Dalíková et al. (2017)

Change made.

Frantisek Marec
16 August 2017

Reviewer #2 (Remarks to the Author):

The revisions look great, especially the new Figure 1.

Reviewer #3 (Remarks to the Author):

The authors have addressed all of my comments satisfactorily. I have no additional comments on the revised draft. This work will be of broad interest to the readers of Nature Communications.